# ICL-NoiseUNet - A Novel In-Context Learning Based Framework For Ultrasound Segmentation With Adaptive Noise Modulation

**Ioannis Charisiadis**[1,3]                                                    I.CHARISIADIS@UVA.NL
**Ilyass el Allali**[1]                                              ILYASS.EL.ALLALI@STUDENT.UVA.NL
**Richard G. P. Lopata**[2]                                                      R.LOPATA@TUE.NL
**Clara I. Sánchez**[1,3]                                        C.I.SANCHEZGUTIERREZ@UVA.NL
**Navchetan Awasthi**[1,3]                                                      N.AWASTHI@UVA.NL

[1] *University of Amsterdam, Faculty of Science, Mathematics and Computer Science, Informatics Institute, Amsterdam, The Netherlands*

[2] *Eindhoven University of Technology, Department of Biomedical Engineering, Eindhoven, The Netherlands*

[3] *Amsterdam UMC, Department of Biomedical Engineering and Physics, Amsterdam, The Netherlands*

**Editors:** Accepted for publication at MIDL 2026

## Abstract

The complex patterns, artifacts and speckle noise that are present in ultrasound images make precise segmentation very challenging. Existing approaches, such as convolutional neural network architectures and foundation models, have shown promising results across a wide range of tasks. However, they struggle to adapt to the unique characteristics of ultrasound data, leading to poor delineation of anatomical boundaries. For that reason, we propose ICL-NoiseUNet, an in-context-learning segmentation framework that combines guidance from a set of input-output pairs, called the context set, with analytic noise descriptors. More specifically, the model leverages an In-Context Feature Conditioning (ICFC) module to incorporate context examples and a Noise Modulation Block (NMB) that adapts feature representation to ultrasound characteristics. After extensive evaluation across several datasets, ICL-NoiseUNet consistently outperforms state-of-the-art methods, enhancing the segmentation quality. Moreover, ablation studies confirm the synergy effect of contextual conditioning and noise modulation. Overall, these findings pave the way for noise-guided ultrasound segmentation. The code will be open-source at https://github.com/johnchart98/ICL-Noise_UNet.git.

**Keywords:** ultrasound segmentation, in-context learning, noise-aware segmentation

## 1. Introduction

Medical image segmentation is one of the most important tasks in medical imaging, as it is applied for clinical assessment, disease diagnosis and treatment. After its rapid development, deep learning has become the predominant approach to tackle this challenge. For example, classical convolutional architectures like U-Net (Ronneberger et al., 2015), W-Net (Xia and Kulis, 2017), nnU-Net,(Isensee et al., 2018) and transformer-based variants like TransUNet (Chen et al., 2021) and UNetTransformer (Hatamizadeh et al., 2021) are widely

considered standard baselines.

Nevertheless, regarding ultrasound imaging, these models often fail to identify critical anatomical details due to ultrasound characteristics such as speckle noise and high contrast variability and acoustic shadowing (Xiao et al., 2025). To enhance generalization across patient populations and imaging conditions, recent studies have explored conditioning segmentation models on a few related examples, referred to as a context set. Methods such as Neuralizer (Czolbe and Dalca, 2023), Universeg (Butoi et al., 2023) and MultiverSeg (Wong et al., 2025) leverage contextual priors from a small number of samples. Furthermore, (Gao et al., 2025) developed an in-context learning framework that encodes reference examples into task-specific embeddings to guide a pretrained segmentation model.

However, previous in-context learning methods continue to struggle in ultrasound applications. These methods consist of target–context fusion modules, where the target and context feature representations are distorted because of ultrasound characteristics. Thus, their segmentation capabilities are limited. At the same time, foundation models such as SAM2 (Ravi et al., 2024), MedSAM (Ma et al., 2024), MEDSAM2 (Ma et al., 2025) and UltraSAM (Meyer et al., 2025) have emerged as universal segmentation frameworks that are capable of generating high-quality masks from simple prompts, like points. While these models have excelled in a wide range of tasks, their performance remains suboptimal in ultrasound segmentation. More specifically, their patch-based architectures cannot mitigate the domain shift between ultrasound data and the general datasets used to train SAM-family models.

Consequently, it is understood that the primary factor limiting the performance of all current approaches is their inability to account for the inherent image characteristics of the ultrasound modality. To address these challenges, we propose **ICL-NoiseUNet**, which introduces a Noise Modulation Block (NMB) mechanism that adaptively refines feature maps based on the ultrasound-specific patterns. Moreover, it employs an In-Context Feature Conditioning (ICFC) module to incorporate context examples and guide feature maps towards expected anatomical structures. Overall, our main contributions can be summarized as follows:

- We introduce a novel context-conditioned segmentation framework that explicitly integrates analytic variance and residual noise maps into the learning process. Thus, our method constitutes a form of noise-aware in-context learning, a subclass that combines contextual guidance with analytic noise modeling to adapt predictions to modality-specific challenges.

- We design a Noise Modulation Block (NMB) that dynamically adapts model architecture to ultrasound statistics.

- We conduct an extensive evaluation across multiple ultrasound segmentation tasks by comparing our method against state-of-the-art context-learning and foundation model architectures.

## 2. Methodology

### 2.1. Overview

ICL-NoiseUNet follows a U-Net–shaped (Ronneberger et al., 2015) design that consists of four encoder blocks, one bottleneck block, four decoder blocks and a segmentation head. Each of these blocks processes the features of the target image through three sequential modules, which are the following:

1. **Feature Extraction Module:** Two consecutive Convolutional ($3 \times 3$ )–Batch Normalization–ReLU layers are utilized to extract feature maps $F_k^t$ at the block $k$ for the target input $x_t$.

2. **Noise Modulation Block (NMB):** Residual and variance noise maps are combined at the block $k$ to form a modulation factor $M_k$, which adapts features $F_k^t$ to ultrasound characteristics. More specifically, the NMB generates modulated feature representations $F_k^{t,mod}$ by suppressing speckle and identifying high-contrast regions. (details in Subsection 2.2).

3. **In-Context Feature Conditioning (ICFC):** Regarding context images, features $F_k^i$ are extracted at each block $k$ in parallel by a separate U-Net encoder–decoder backbone (without NMB or ICFC) that shares weights across all context images $x_i$. Then, refined target features $F_k^{t,mod}$ are fused with context features $F_k^i$ via channel concatenation, followed by shared $1 \times 1$ convolution and average pooling to obtain context-informed representations $\hat{F}_k^t$ (see Subsection 2.3). In Figure 1, it is stated as Target-Context Fusion Block. In this way, the context set guides the model towards expected anatomical structures. This target-context fusion mechanism draws inspiration from prior In-Context-Learning designs such as Neuralizer (Czolbe and Dalca, 2023).

At the end of the bottleneck and decoder blocks, transpose convolutional layers are applied to restore spatial resolution. Also, skip connections link the encoder and decoder stages. Finally, the output mask $\hat{y}_t$ is generated by the segmentation head, which consists of a $1 \times 1$ convolution layer followed by a sigmoid activation function. An overview of the complete architecture is presented in **Figure 1**.

The main role of the NMB is to reduce the effect of ultrasound speckle on feature activations. It refines feature representations using residual and variance noise maps and acts as a noise-aware normalization mechanism across the network. Thus, it reduces the effect of noise-related variations and ensures that features representing similar anatomical structures are more comparable, even when noise patterns or scanner settings vary. As a result, context conditioning operates in a more semantically meaningful feature space, allowing contextual information to guide segmentation more effectively. Without NMB, noise variability can propagate through the network and weaken context alignment. Therefore, NMB complements rather than replaces contextual guidance. Additionally, unlike all previous in-context learning approaches that rely solely on a sequence of target–context fusion blocks, we introduce multi-stage feature extraction before the fusion stage. This design helps the model to capture richer, more informative feature representations that interact effectively, increasing its capabilities.

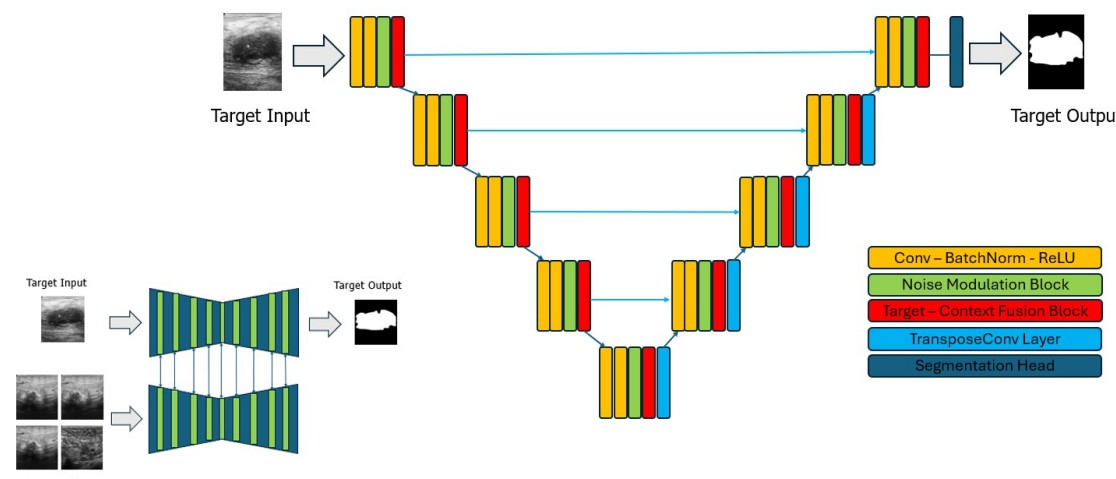

Figure 1: ICL-NoiseUNet follows a UNet-shaped (Ronneberger et al., 2015) en-
coder–decoder design that integrates noise modulation and target–context fu-
sion modules as core components. Context examples guide segmentation through
target-context fusion modules, providing structural priors that enhance the target
representation (down left).

## 2.2. Noise Modulation Block for Ultrasound Robustness

To enhance the quality of ultrasound segmentation, two complementary maps are utilized:
(i) a residual noise map, computed as the difference between the image and its Gaussian-
smoothed version. and (ii) a local variance map, which measures how much the intensity
fluctuates within a small neighborhood. The variance map highlights regions dominated by
speckle, where feature activations should be suppressed. On the other hand, the residual
noise map captures details, such as anatomical edges, that should be preserved. Combining
both, we provide a balanced representation of ultrasound patterns. Thus, we integrate
**NMB** at each encoder and decoder level, as seen in Figure 1. Two analytic descriptors
are utilized for our Noise Modulation Block: a residual noise map

$$n_r(x_t) = |x_t - (G_\sigma * x_t)|, \tag{1}$$

where $G_\sigma$ denotes a Gaussian smoothing kernel. The second one is a local variance map,
which identifies regions with high variability.

$$n_v(x_t) = \mathbb{E}[x_t^2] - (\mathbb{E}[x_t])^2, \tag{2}$$

These maps are precomputed before the forward pass for each image during training and
inference. At each encoder–decoder level, they are resized and combined to form the mod-
ulation factor $M_k$ with the following equation:

$$M_k = 1 + \alpha_k n_r(x_t) + \beta_k n_v(x_t), \tag{3}$$

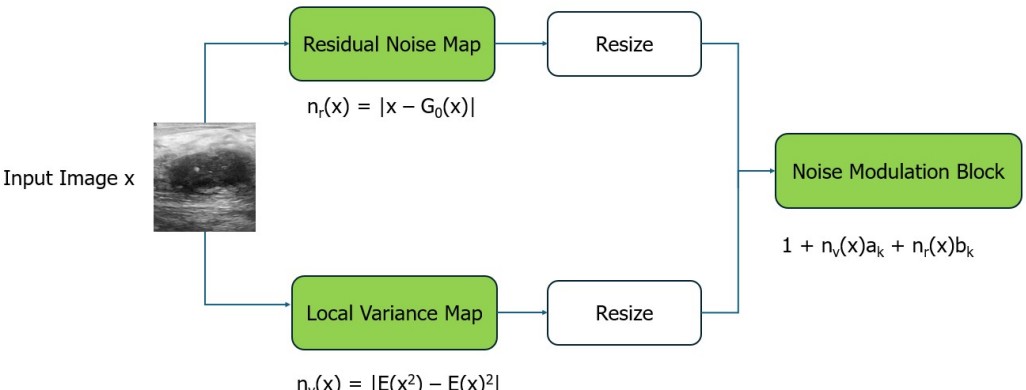

Figure 2: Noise Modulation Block (NMB). Residual and variance maps modulate activations through learnable weights, improving boundary preservation under speckle noise.

where $a_k$ and $b_k$ are learnable scalar parameters. Consequently, the feature map $F_k^t$ is then refined as

$$F_k^{t,\mathrm{mod}} = F_k^t \odot M_k. \tag{4}$$

The process is depicted in **Figure 2**. In this way, the target feature representations become semantically richer and less affected by speckle. Thus, their interaction with context features in ICFC module is more reliable.

### 2.3. In-Context Feature Conditioning (ICFC) Module

Context images are encoded using the exact and context feature maps $F_k^i$ are generated. To fuse information from context feature maps, we perform channel concatenation of each context feature with the noise-modulated feature map of the input image:

$$f_k^{c,i} = [F_k^{t,\mathrm{mod}} \| F_k^i], \tag{5}$$

Then, the concatenated features $f_k^{c,i}$ are passed through a shared $1 \times 1$ convolution.

$$\tilde{f}_k^{c,i} = \phi_k(f_k^{c,i}). \tag{6}$$

Subsequently, pairwise contextual features, stated as $\tilde{f}_k^{c,i}$, are aggregated via mean pooling. Finally, a residual refinement with our initial target features $F_k^t$ is applied to update the target feature map:

$$\hat{F}_k^t = \mathrm{GeLU}\big(F_k^{t,\mathrm{mod}} + \mathrm{Mean}_i(\tilde{f}_k^{c,i})\big), \tag{7}$$

The mechanism of target-context information is illustrated in Figure 3.

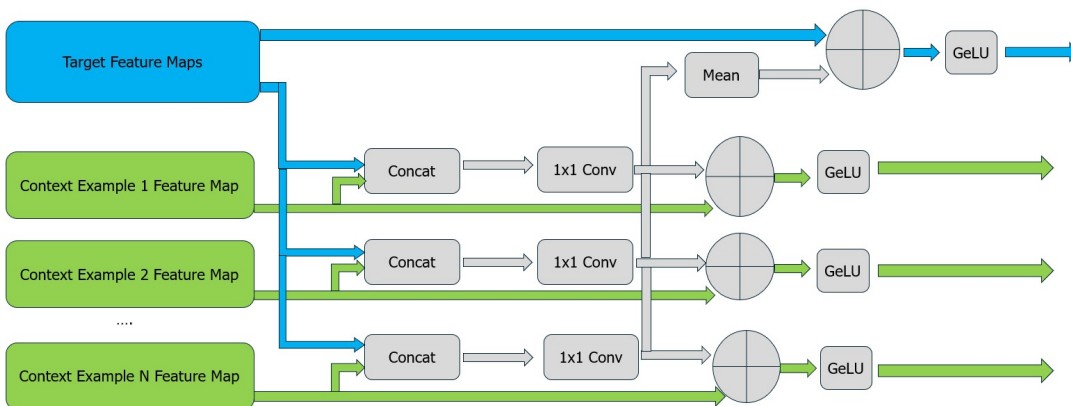

Figure 3: Target-Context fusion block. The target feature map (blue) is concatenated with each context feature map (green).

## 3. Experiments and Results

We design our experiments to test whether the combination of two key ideas of ICL-NoiseUNet, the feature refinement by the NMB and the conditioning of the segmentation model from the context set, improves ultrasound segmentation. Because of the unique properties of the ultrasound modality, we hypothesize that NMB should stabilize early representations, while context examples should guide the model towards plausible anatomical priors. Thus, we evaluate: (i) the contribution of each component to the model's capability (ablations), (ii) the model's performance against state-of-the-art models, (iii) the effect of context size selection and (iv) cross-dataset performance robustness within the same task. More specifically, we test **ICL-NoiseUNet** across 4 ultrasound segmentation tasks: fetal head, breast lesion, thyroid gland and cardiac chamber. Furthermore, information regarding the publicly available used datasets: RADBOUD (van den Heuvel et al., 2018), JNU (Lu et al., 2022), BUSI (Al-Dhabyani et al., 2020), BUS-BRA (Wilfrido Gómez-Flores et al., 2023)), Magdeburg Thyroid (Wunderling et al., 2017) and TG3K (Gong et al., 2022) can be found in Appendix A. For all datasets, splits are performed at a patient level, ensuring that no patient appears in more than one split and preventing leakage of highly similar frames across training, validation, and test sets. Additionally, context is retrieved only from the training data, and all patient identities are completely separate from those used in evaluation and testing. For video-based datasets (JNU-IFM (Lu et al., 2022) and CAMUS (Leclerc et al., 2019)), all frames from a given video are assigned to a single split. For BUSI (Al-Dhabyani et al., 2020) and BUS-BRA (Wilfrido Gómez-Flores et al., 2023), patient-level splits are additionally stratified by pathology; for BUS-BRA, we also report results stratified by scanner to support the noise/statistics adaptation analysis in Appendix I I. We train using PyTorch Lightning for 50 epochs with the AdamW optimizer (lr=$1 \times 10^{-5}$). Regarding the window size for residual and variance maps, it is set to 7 (further analysis at Appendix C). During training, context samples are randomly drawn from the training set with a fixed size of $L = 4$ (see Section 3.3 for explanation). At inference time, the 4

context images are selected from the training set using the lowest $L_2$ distance to the target image. We use L2 as it is fast to compute, deterministic and does not require additional parameters, unlike perceptual metrics such as SSIM (Wang et al., 2004) or LPIPS (Zhang et al., 2018). Full implementation details, augmentation transformations, data splits and hardware specifications are provided in Appendix B.

### 3.1. Ablation Study for Context Conditioning and Noise Modulation Block

Table 1 presents the ablation experiments on BUSI (Al-Dhabyani et al., 2020), BUS-BRA (Wilfrido Gómez-Flores et al., 2023), and RADBOUD (van den Heuvel et al., 2018) datasets to analyze the contribution of contextual conditioning and NMB. We observe that by removing either component reduces Dice and IoU, confirming that both mechanisms provide complementary benefits. On BUSI (Al-Dhabyani et al., 2020), Dice improves from 0.74 to 0.80 when both modules are active, while on BUS-BRA (Wilfrido Gómez-Flores et al., 2023), the full ICL-NoiseUNet achieves 0.93 Dice and 0.87 IoU, representing an improvement of over 3% compared to single-component variants. Similarly, on RADBOUD (van den Heuvel et al., 2018), our model achieves a Dice of 0.96, demonstrating a significant gain in segmentation stability over the reduced variants.

All differences relative to the full model are statistically significant, so they validate our hypothesis. A detailed analysis of the complementary effect of variance and residual noise maps that form the NMB can be found in Appendix C. Additionally, the synergy between context and noise maps is also evaluated when using shared encoder–decoder weights for the target and context (see Appendix E), showing only a minimal change in performance. In addition, an analysis of the learned modulation parameters ($\alpha_k$ and $\beta_k$) across network depth and datasets is provided in Appendix K. More specifically, it offers further insights into how the model adapts noise handling to different tasks.

Table 1: Ablation study on the effect of noise modulation block (nm) and context conditioning (cc) across multiple ultrasound segmentation datasets. Each metric's score is reported along with its standard deviation across 5 runs with different random seeds. Statistical significance was assessed using Wilcoxon signed-rank tests. Significance levels: † $p < 0.05$, ‡ $p < 0.01$, § $p < 0.001$.

| Model Variant | BUSI | | BUS-BRA | | RADBOUD | |
|---|---|---|---|---|---|---|
| | Dice | IoU | Dice | IoU | Dice | IoU |
| ICL-NoiseUNet | **0.804 ± 0.043** | **0.702 ± 0.058** | **0.931 ± 0.013** | **0.877 ± 0.022** | **0.961 ± 0.010** | **0.947 ± 0.013** |
| ICL-NoiseUNet (-nm) | 0.751 ± 0.051 | 0.632 ± 0.061 | 0.891 ± 0.019 | 0.848 ± 0.025 | 0.932 ± 0.011 | 0.908 ± 0.015 |
| ICL-NoiseUNet (-nm,-cc) | 0.741 ± 0.155 | 0.657 ± 0.186 | 0.871 ± 0.016 | 0.818 ± 0.019 | 0.928 ± 0.093 | 0.875 ± 0.112 |
| **Wilcoxon Signed-Rank Pairwise Test Results for Dice Scores** | | | | | | |
| BUSI | Full vs (-nm): † ; Full vs (-nm,-cc): ‡; (-nm) vs (-nm,-cc): $p = 0.183$ | | | | | |
| BUS-BRA | Full vs (-nm): §; Full vs (-nm,-cc): §; (-nm) vs (-nm,-cc): § | | | | | |
| RADBOUD | Full vs (-nm): §; Full vs (-nm,-cc): §; (-nm) vs (-nm,-cc): § | | | | | |

**Sensitivity Analysis of the Noise Modulation Block** To evaluate the contribution of each of the two components of the NMB, we perform an ablation study on the CAMUS (Leclerc et al., 2019) and BUS–BRA (Wilfrido Gómez-Flores et al., 2023) datasets. We compare four variants: (1) Full NMB (residual + variance maps), (2) Residual noise–only computation, (3) Variance–only computation, and (4) No NMB. As shown in Table 2, the

full design that combines the residual and variance maps achieves the strongest results, whereas single-component variants perform consistently worse and the removal of the entire NMB leads to the largest performance drop. Additionally, all differences relative to the full model are statistically significant which confirms their complementary effect. Moreover, this argument is also enhanced by qualitative examples shown in Appendix C. There, it is shown that the residual-only variant preserves sharp boundaries but often misses low-contrast or blurred regions, resulting in increased false negatives. In contrast, the variance-only variant suppresses speckle but frequently overextends into surrounding tissue, leading to a higher number of false positives. By combining both descriptors, the full block results in more anatomically consistent segmentation.

**Feature Representation Analysis** To further analyze the impact of NMB, we visualize bottleneck feature representations using t-SNE on 100 BUS-BRA (Wilfrido Gómez-Flores et al., 2023) samples (33 benign, 67 malignant) (Fig. 9, Appendix H). Without NMB, benign and malignant features are poorly separated, indicating weak class awareness in the latent space. In contrast, incorporating NMB results in more compact and class-aligned clusters with significantly reduced overlap, which is also reflected by a much higher silhouette score (0.26 vs. 0.05). Although t-SNE is primarily a visualization tool, the improved separability suggests that NMB promotes more structured and discriminative feature representations. Consequently, this improved representation structure helps explain the observed gains in segmentation performance.

Table 2: Ablation study of the Noise Modulation Block (NMB). We report mean Dice and IoU scores (mean ± std) across 5 runs. Reported $p$–values correspond to Wilcoxon signed-rank tests comparing each variant with the Full NMB model. Significance levels: † $p < 10^{-5}$

| Model Variant | CAMUS | | | BUS–BRA | | |
|---|---|---|---|---|---|---|
| | Dice ↑ | IoU ↑ | $p$–value | Dice ↑ | IoU ↑ | $p$–value |
| **Full NMB (Residual + Variance)** | **0.940 ± 0.008** | **0.891 ± 0.012** | – | **0.931 ± 0.013** | **0.877 ± 0.022** | – |
| Residual Only | 0.926 ± 0.009 | 0.869 ± 0.011 | † | 0.881 ± 0.014 | 0.829 ± 0.023 | † |
| Variance Only | 0.929 ± 0.010 | 0.871 ± 0.011 | † | 0.886 ± 0.014 | 0.834 ± 0.021 | † |
| No NMB | 0.921 ± 0.011 | 0.867 ± 0.013 | † | 0.891 ± 0.019 | 0.848 ± 0.025 | † |

## 3.2. Comparison with SOTA Models

For fair comparison, the evaluated models are trained under an identical experimental setup. In particular, we follow identical dataset splits, data augmentations, training and inference settings. Furthermore, in datasets that containing multiple classes, we ensure that each split preserves the class proportions of the full dataset. Finally, for SAM-based models, we provide positive points as prompts. More specifically, for SAM-family models, we conducted 5 independent runs per test image. At each run, we use 5 positive point prompts sampled from within the ground truth foreground mask: 4 points close to the spatial extremes (xmin, xmax, ymin, ymax of the mask bounding region) to provide boundary coverage, plus 1 additional point sampled randomly from the interior region. This strategy provides SAM with good spatial coverage while introducing variability across runs through the randomly

sampled points. Table 3 compares ICL-NoiseUNet with CNN-based and foundational models on the BUS-BRA (Wilfrido Gómez-Flores et al., 2023) dataset. The proposed model achieves a Dice score of 93%, beating UltraSAM (Meyer et al., 2025) by 8% and MedSAM2 (Ma et al., 2025) by 13%. These results highlight the benefits of combining contextual conditioning with analytic noise modulation. To further assess our model's robustness, we evaluate it additionally on CAMUS (Leclerc et al., 2019) and BUSI (Al-Dhabyani et al., 2020) datasets. Consequently, as reported in Table 3, our method reaches Dice scores of 0.94 on CAMUS (Leclerc et al., 2019), 0.80 on BUSI (Al-Dhabyani et al., 2020) and 0.96 on RADBOUD (van den Heuvel et al., 2018), outperforming a range of competitors, including SwinUNet (Cao et al., 2021), nnU-Net (Isensee et al., 2018), MedSAM2 (Ma et al., 2025) and the most recent in-context learning model, MultiverSeg (Wong et al., 2025). .

Table 3: Comparison with current State-of-The-Art models across diverse medical imaging datasets.Significance levels: † $p < 0.05$, ‡ $p < 0.01$, § $p < 0.001$. We compute the $p$-value for Dice score significance against ICL-NoiseUNet using Wilcoxon signed-rank pairwise tests.

| Model | CAMUS | | BUSI | | BUS-BRA | | RADBOUD | |
|---|---|---|---|---|---|---|---|---|
| | Dice | Sig. | Dice | Sig. | Dice | Sig. | Dice | Sig. |
| UNEt TRansformers (Hatamizadeh et al., 2021) | 0.921 | § | 0.761 | † | 0.923 | † | 0.881 | § |
| SwinUNet (Cao et al., 2021) | 0.912 | § | 0.642 | § | 0.911 | ‡ | 0.900 | § |
| nnU-Net (Isensee et al., 2018) | 0.915 | § | 0.742 | § | 0.921 | † | 0.921 | § |
| MultiverSeg (Wong et al., 2025) | 0.882 | § | 0.643 | § | 0.781 | § | 0.901 | § |
| UltraSAM (Meyer et al., 2025) | 0.901 | ‡ | 0.746 | ‡ | 0.855 | § | 0.935 | ‡ |
| MedSAM2 (Ma et al., 2025) | 0.892 | § | 0.751 | ‡ | 0.794 | § | 0.940 | ‡ |
| ICL-NoiseUNet (ours) | **0.940** | – | **0.804** | – | **0.931** | – | **0.961** | – |

We conducted further experiments comparing against established few-shot segmentation techniques (PANet (Wang et al., 2020) and feature-wise conditioning mechanisms that are applied to our noise maps (FiLM (Perez et al., 2017) and Conditional BatchNorm (de Vries et al., 2017)). ICL-NoiseUNet consistently outperforms all baselines on both CAMUS (Leclerc et al., 2019) and scanner-stratified BUS-BRA (Wilfrido Gómez-Flores et al., 2023) (detailed results in Appendix J). It is demonstrated that PANet struggles in ultrasound segmentation, while generic feature-wise conditioning methods are less effective than our NMB design.

## 3.3. Effect of Context Size on Segmentation Performance

To assess the impact of context size on our model's performance, we trained ICL-NoiseUNet with different context sizes of ($L = 1, 2, 4, 8, 16$) and measured Dice coefficient scores across the BUS-BRA (Wilfrido Gómez-Flores et al., 2023), CAMUS (Leclerc et al., 2019) and RADBOUD (van den Heuvel et al., 2018) datasets. Figure 1 5 shows the effect of context size in segmentation performance across each dataset (more detailed results for BUS-BRA (Wilfrido Gómez-Flores et al., 2023) are reported in Appendix D). There, we observe a slight but consistent improvement as the context size increases from $L = 1$ to $L = 4$. As the size increases further, the performance stops improving and gets noticeably worse, especially in the BUS-BRA (Wilfrido Gómez-Flores et al., 2023) dataset. This happens because

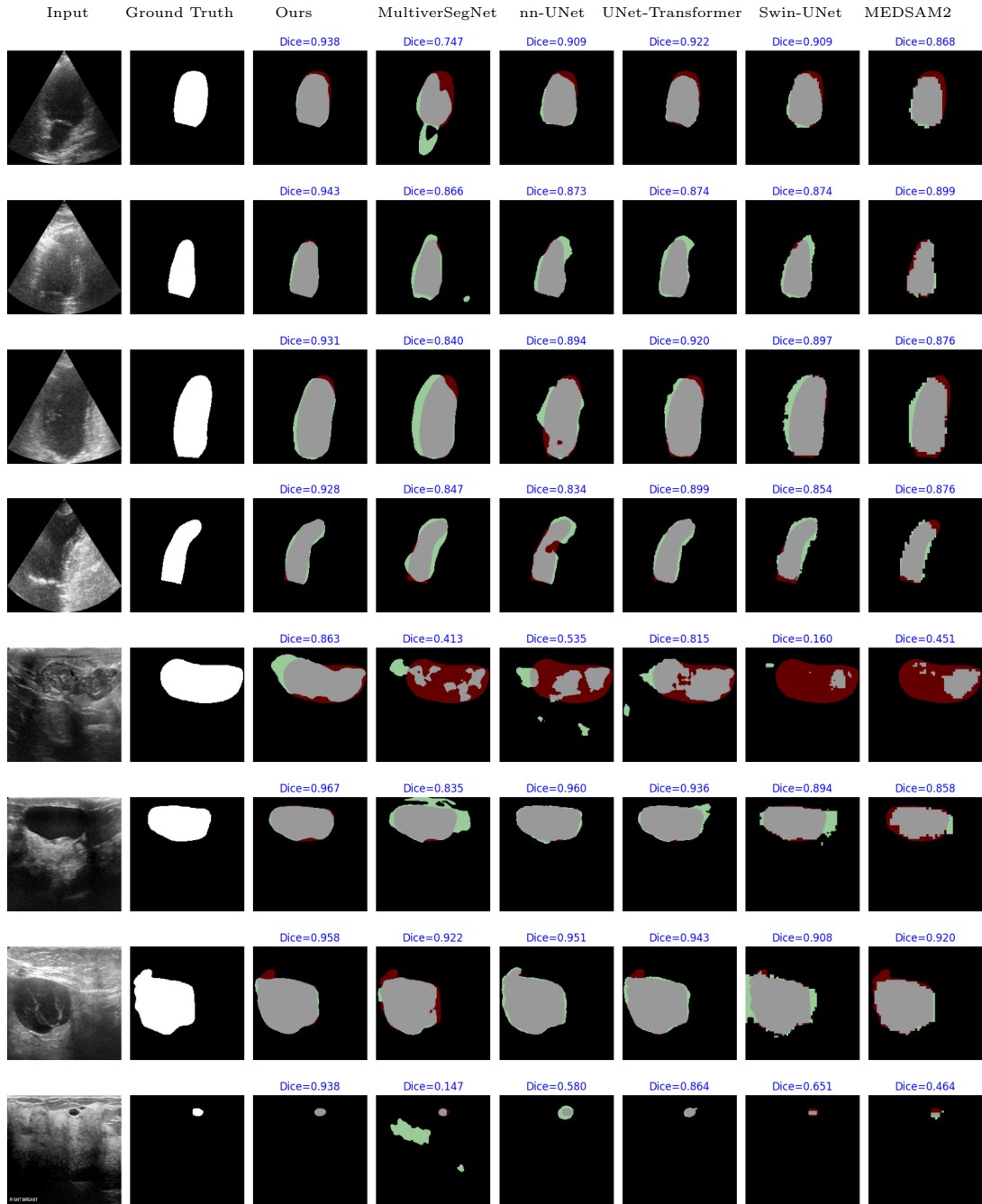

Figure 4: Segmentation results comparison between the proposed *ICL-NoiseUNet* and other state-of-the-art models on samples from CAMUS (Leclerc et al., 2019) (first 4 rows) and BUSI (Al-Dhabyani et al., 2020) (last 4 rows) datasets. Dice score and Intersection-over-Union are reported for each method. ■ indicates false positives and ■ indicates false negatives.

larger context sizes may introduce redundant or less relevant examples whose distribution differs significantly from the target image. Therefore, the effectiveness of context guidance is reduced. The same pattern is observed across all evaluated segmentation tasks, with $L = 4$ achieving the best trade-off between contextual diversity and computational efficiency. Thus, we select $L = 4$ for all main experiments. In contrast to methods such as Neuralizer (Czolbe and Dalca, 2023) and Universeg (Butoi et al., 2023), which rely solely on contextual information, our framework benefits from the synergy between the base encoder–decoder backbone, analytic noise descriptors and context conditioning. In addition, we evaluate our model's sensitivity to the context selection strategy by comparing retrieval based on L2 distance, SSIM index, and random selection. For random selection, we draw the context set from a 10% subsample of the training set. Results at Appendix I indicate minimal differences at BUS-BRA (Wilfrido Gómez-Flores et al., 2023) (0.911 vs 0.902 vs 0.910) and CAMUS (Leclerc et al., 2019) (0.940 vs 0.937 vs 0.936) datasets, respectively. Consequently, these results confirm our model's robustness to the context selection method and that it does not require precise similarity matching. Moreover, the analysis in Appendix G confirms the model's robustness to context size variations during inference.

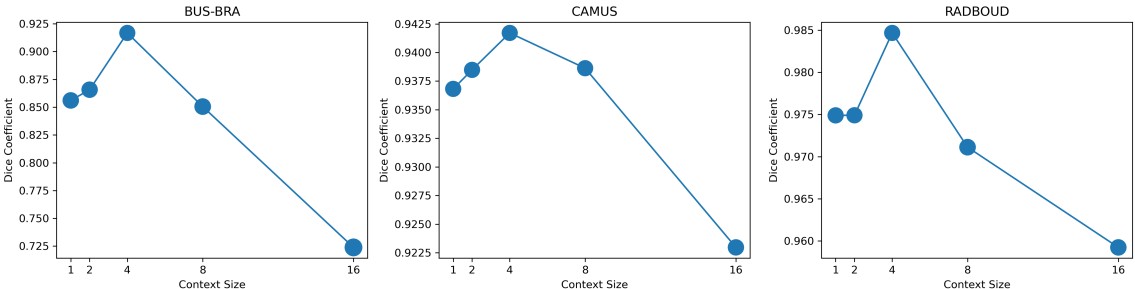

Figure 5: Effect of context size on segmentation performance for the BUS-BRA (Wilfrido Gómez-Flores et al., 2023), CAMUS (Leclerc et al., 2019) and RADBOUD (van den Heuvel et al., 2018) datasets. The marker size is proportional to the Dice standard deviation.

### 3.4. Cross-domain evaluation

To assess domain generalization, Table 4 reports cross-dataset results in fetal head and thyroid gland segmentation. We train the model on JNU-IFM (Lu et al., 2022) and Thyroid-Magdeburg (Wunderling et al., 2017) and test directly on RADBOUD (van den Heuvel et al., 2018) and TG3K (Gong et al., 2022). It is shown that ICL-NoiseUNet achieves Dice scores of 0.901 and 0.921 on the respective test datasets, outperforming baselines that were trained entirely on the RADBOUD (van den Heuvel et al., 2018). The results confirm that noise modulation enhances the model's ability to generalize across datasets of the same task without the need for retraining. Additionally, we compare ICL-NoiseUNet with MixStyle (Zhou et al., 2021), a style-based feature augmentation method, which performs slightly worse in these cross-dataset settings (0.901 vs. 0.911; 0.921 vs. 0.911). This highlights

that explicitly modeling ultrasound noise provides stronger generalization than style-based augmentation alone.

Table 4: Cross-dataset results on fetal head and thyroid gland (Wunderling et al., 2017) segmentation. NMB: noise modulation

| Model / Setting | Dice | IoU | Recall |
|---|---|---|---|
| **JNU-IFM → RADBOUD** | | | |
| ICL-NoiseUNet | **0.901** | **0.887** | **0.972** |
| ICL-NoiseUNet (-NMB) | 0.881 | 0.842 | 0.951 |
| MixStyle (Zhou et al., 2021) | 0.911 | 0.899 | 0.907 |
| **Thyroid-Magdeburg → TG3K** | | | |
| ICL-NoiseUNet | **0.921** | **0.868** | 0.919 |
| ICL-NoiseUNet (-NMB) | 0.870 | 0.811 | 0.900 |
| ICL-NoiseUNet (context from Thyroid) | 0.920 | 0.910 | **0.920** |
| MixStyle (Zhou et al., 2021) | 0.911 | 0.899 | 0.907 |

### 3.5. Limitations and Future Work

Overall, ICL-NoiseUNet demonstrates strong generalization, stability, and robustness across several evaluated datasets. The model's ability to preserve anatomical detail and suppress speckle noise variations shows that it is a practical and interpretable architecture for ultrasound image segmentation. Importantly, the proposed framework is not restricted to a U-Net (Ronneberger et al., 2015) backbone. The NMB and ICFC components are modular and can be integrated into other segmentation architectures. For transformer-based models, the NMB could be integrated after each transformer block and refine the output features with resolution-matched noise maps, while ICFC can be applied by fusing target and context features at each transformer block output. For more lightweight convolutional approaches, NMB can be placed after each convolutional block in the encoder and decoder, with ICFC applied at the corresponding feature levels. Therefore, we plan to evaluate their effectiveness within transformer-based and hierarchical medical imaging models. Although ICL-NoiseUNet achieves strong segmentation performance, it remains a relatively heavyweight model, with a higher number of parameters compared to baseline architectures. Nevertheless, its inference time is still comparable to other models, as summarized in Appendix F.

Future work will also focus on developing more parameter-efficient architectures that combine noise maps with contextual information. Regarding failure cases, we identify two main limitations of our method. Firstly, performance degrades under extreme speckle noise and severe acoustic shadowing, such as in some samples of the BUSI dataset (Al-Dhabyani et al., 2020), where anatomical boundaries are heavily obscured. In these cases, noise overwhelms the underlying anatomical signal to the extent that even context-guided feature refinement cannot reliably recover boundary information. Secondly, our in-context learning approach is task-specific and requires anatomically relevant context examples. Using context images from a different anatomical task (e.g., cardiac contexts for breast lesion segmentation) provides no meaningful guidance, leading to a significant drop in performance due to incompatible structural priors across anatomical domains.

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

## Appendix A. Datasets

### A.1. Fetal Head Segmentation Datasets

During evaluation of our proposed method, we include 2 different datasets for fetal head segmentation.

- The RADBOUD (van den Heuvel et al., 2018)-FetalHC dataset consists of 999 2D ultrasound images of fetal head circumference (HC) measurements, collected from 551 pregnant women during routine screenings at RADBOUD University Medical Center between $2014 - 2015$. The images, sized $800 \times 540$ pixels with a pixel resolution between $0.052 - 0.326$ mm, exclusively include fetuses without growth abnormalities.

- The JNU-IFM (Lu et al., 2022) The dataset comprises 6224 fetal head ultrasound images collected from 78 videos recorded from 51 pregnant women between 2019 and 2020. It originates from the Intelligent Fetal Monitoring Lab of Jinan University. It is categorized into four labels: *None* (1,022 images), *OnlySP* (323 images), *Only-Head* (1,136 images), and *SPHead* (3,743 images). We evaluate only on samples with *OnlyHead* label

### A.2. Thyroid Gland Datasets

For thyroid analysis, two datasets were used.

- The Magdeburg-Thyroid3D (Wunderling et al., 2017) thyroid dataset includes sixteen 3D ultrasound scans of healthy thyroid lobes from the University Hospital Magdeburg[1], each with expert-annotated segmentations. To utilize the dataset, we split 3D into 2D slices and we kept only the slices containing thyroid annotations. Thus, we obtain 2D image–mask pairs which are used for our experiments.

- The TG3K (Gong et al., 2022) dataset comprises of 3585 2D B-mode thyroid gland ultrasound images acquired via GE Logiq E9 scanners, with pixel-wise annotations for gland segmentation .

### A.3. CAMUS (Cardiac Acquisitions for Multi-structure Ultrasound Segmentation) Dataset

The CAMUS dataset consists of 2D echocardiographic image sequences of apical four-chamber and two-chamber views. They are acquired from 500 patients at the University Hospital of Saint-Étienne (France) (Leclerc et al., 2019).

- Images include expert annotations of the left ventricular endocardium, epicardium, and left atrium at end-diastole and end-systole. Thus, they encompass a wide range of cardiac pathologies and imaging conditions.

- The dataset is publicly available for benchmarking cardiac segmentation and volume estimation tasks in echocardiography.

---

1. https://www.med.uni-magdeburg.de/en/

## A.4. BUSI (Breast Ultrasound Images Dataset) Dataset

The BUSI dataset (Al-Dhabyani et al., 2020) includes a collection of B-mode breast ultrasound images used for lesion segmentation and classification.

- The dataset comprises of 780 ultrasound images from approximately 600 female patients aged between $25 - 75$ years. The images are captured using LOGIQ E9 and LOGIQ E9 Agile ultrasound systems at Baheya Hospital in Egypt.

- Each image, which has a resolution of $500 \times 500$ pixels, is categorized into three patient groups: normal (133), benign (437), and malignant (210).

- For benign and malignant cases, pixel-level segmentation masks of the lesion areas are provided to facilitate tumor delineation and carry out further analysis.

## A.5. BUS-BRA (Breast Ultrasound Dataset)

The BUS-BRA dataset (Wilfrido Gómez-Flores et al., 2023) provides a large-scale breast ultrasound image collection designed to support computer-aided diagnosis and segmentation evaluation.

- It contains 1875 anonymized ultrasound images from 1064 patients undergoing routine breast ultrasound examinations in Brazil. Moreover, they acquired using four different scanners.

- Each image includes biopsy-proven lesion annotations and two labels (benign and malignant), along with expert-annotated segmentation masks of the lesion and breast tissue regions.

# Appendix B. Implementation Details For ICL-NoiseUNet

Table 5: Implementation details used for training ICL-NoiseUNet.

| Component | Setting |
| --- | --- |
| Framework | PyTorch Lightning |
| Optimizer | AdamW |
| Learning rate | $1 \times 10^{-5}$ |
| Weight decay | $1 \times 10^{-7}$ |
| Batch size | 4 (1 target + 4 context samples) |
| Epochs | 50 |
| Early stopping | Patience = 8 epochs (validation loss) |
| Loss function | Dice Loss |
| Context size (train) | $L = 4$ (random sampling) |
| Context selection (inference) | 4 images with the minimum L2 distance |
| Noise map window size | $7 \times 7$ |
| Augmentation (targets only) | Flips, rotation, elastic deformation, zoom, Gaussian noise, brightness/contrast |
| Context augmentation | None |
| Dataset split | 70 / 15 / 15 (train/validation/test) |
| GPU | NVIDIA GeForce GTX 1080 (1 GPU) |
| Checkpointing | Best model via validation loss |

# Appendix C. Sensitivity Analysis of the Noise Modulation Block

## C.1. Qualitative Results

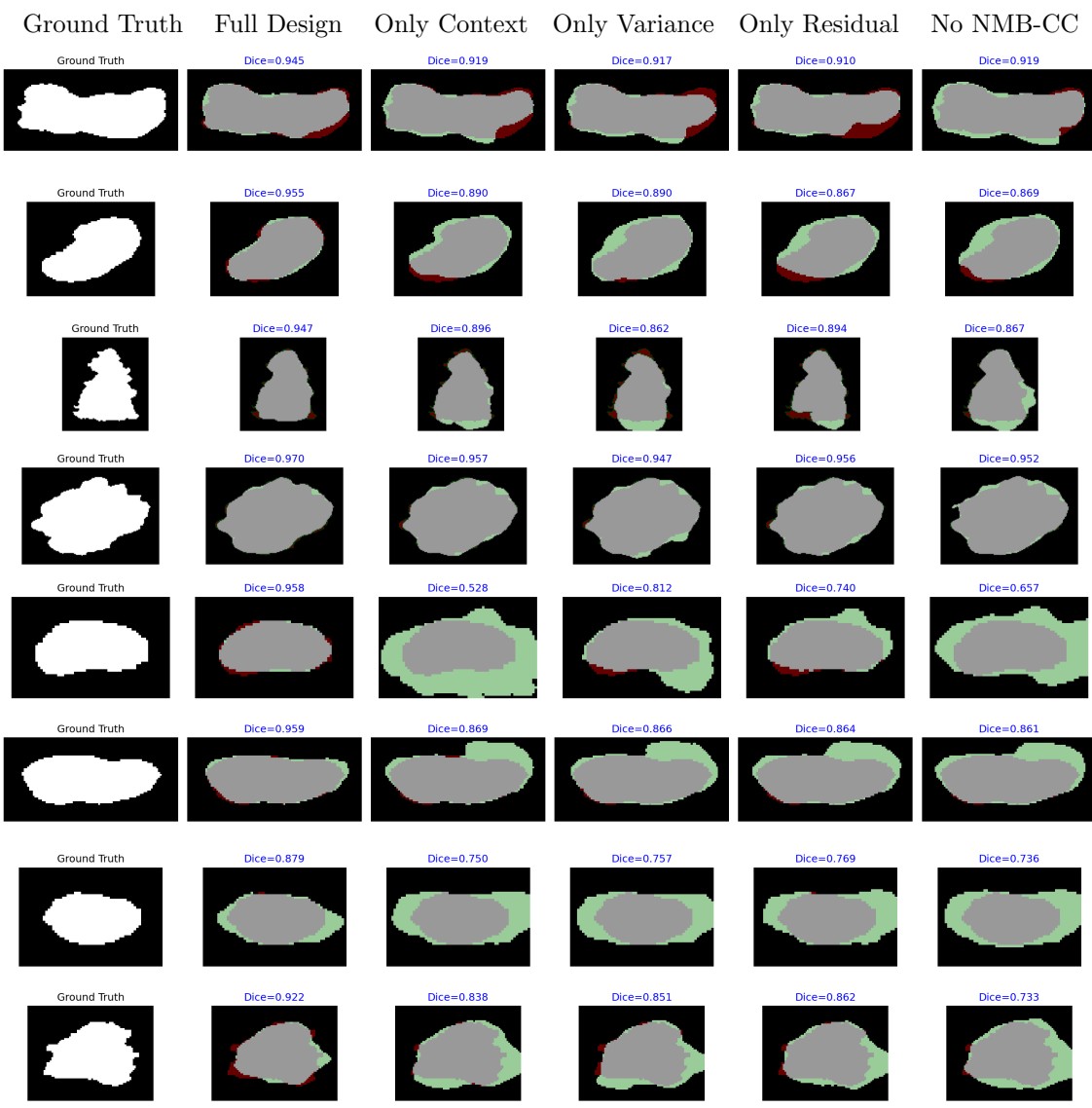

Figure 6: Segmentation results comparison between variations of our approach for BUS-BRA (Wilfrido Gómez-Flores et al., 2023) dataset. For clearer visualization, images are cropped around the region of interest. ▉ indicates false positives, and ▉ indicates false negatives. Abbreviations: noise modulation block (NMB), context conditioning (CC)

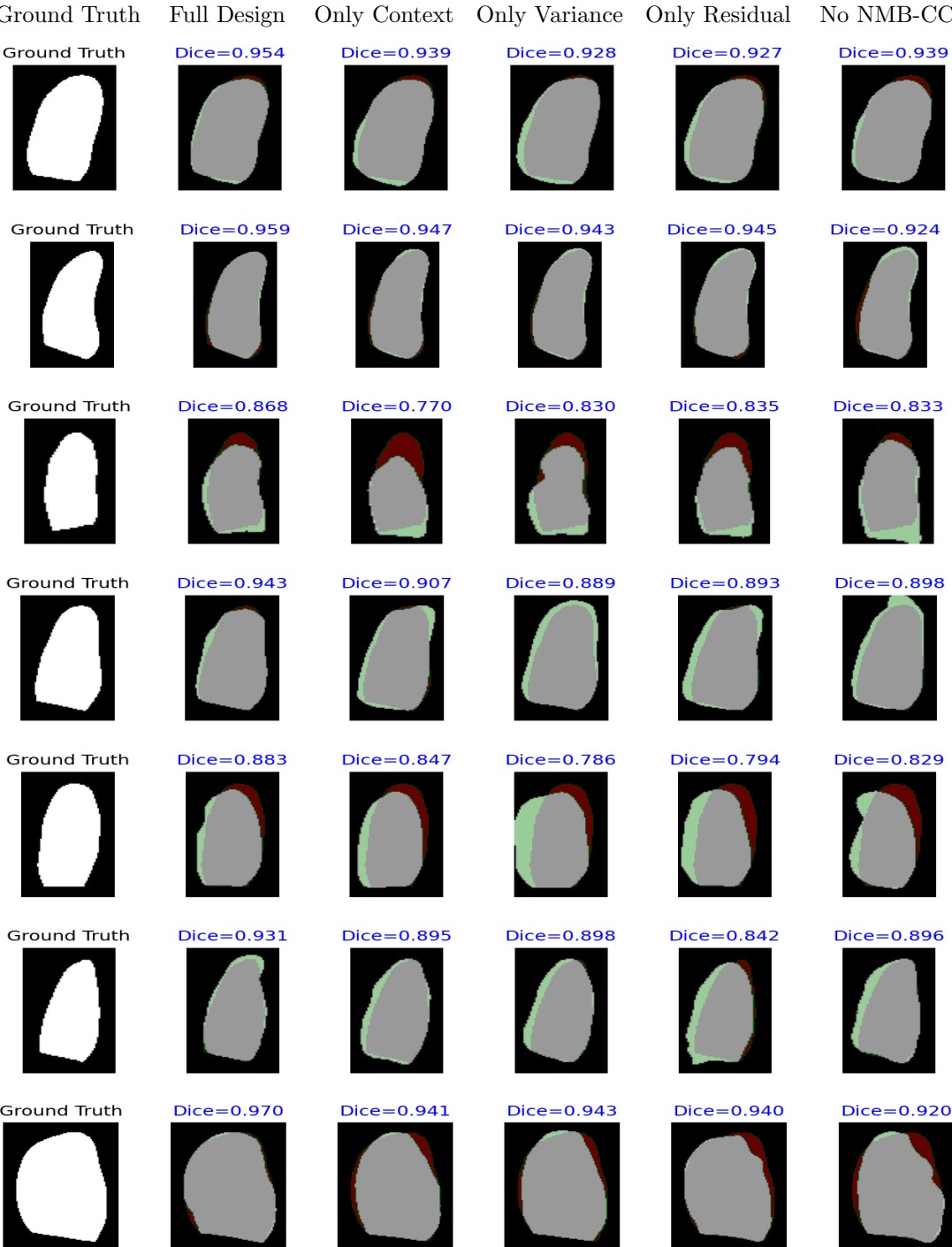

Figure 7: Segmentation results comparison between variations of our approach for CAMUS (Leclerc et al., 2019) dataset. For clearer visualization, images are cropped around the region of interest. ■ indicates false positives, and ■ indicates false negatives. Abbreviations: noise modulation block (NMB), context conditioning (CC)

### C.2. Window Size Selection For Residual Noise and Variance Maps

Table 6 shows that a window size of 7 gives the most reliable noise maps for the model. Smaller windows (3 and 5) extract little local context, leading to weaker noise estimation and lower accuracy. A larger window (9) fails to capture important boundaries. Thus, window size 7 provides the best balance between detail and context, which is reflected in the highest Dice and IoU scores across both datasets. Therefore, we select 7 as the default window size in our model.

Table 6: Ablation study of the window size for our residual and variance noise maps. Reported $p$–values correspond to Wilcoxon signed–rank tests comparing each variant with the Full NMB model.

| Window Size | CAMUS | | | BUS–BRA | | |
|---|---|---|---|---|---|---|
| | Dice ↑ | IoU ↑ | $p$–value | Dice ↑ | IoU ↑ | $p$–value |
| Window Size = 3 | $0.929 \pm 0.010$ | $0.872 \pm 0.012$ | $p = 2 \times 10^{-9}$ | $0.871 \pm 0.014$ | $0.819 \pm 0.023$ | $p < 10^{-20}$ |
| Window Size = 5 | $0.932 \pm 0.011$ | $0.876 \pm 0.011$ | $p = 1.5 \times 10^{-6}$ | $0.875 \pm 0.015$ | $0.803 \pm 0.022$ | $p < 10^{-20}$ |
| Window Size = 7 | $\mathbf{0.940 \pm 0.008}$ | $\mathbf{0.891 \pm 0.012}$ | – | $\mathbf{0.931 \pm 0.013}$ | $\mathbf{0.877 \pm 0.022}$ | – |
| Window Size = 9 | $0.934 \pm 0.009$ | $0.878 \pm 0.012$ | $p = 1 \times 10^{-6}$ | $0.883 \pm 0.014$ | $0.832 \pm 0.023$ | $p < 10^{-20}$ |

## Appendix D. Additional Results Regarding The Effect of Context Size on Segmentation Performance

Table 7: Segmentation performance of ICL-NoiseUNet with varying context sizes ($L$) on BUS-BRA (Wilfrido Gómez-Flores et al., 2023) dataset. Mean $\pm$ standard deviation reported for Dice and IoU. Lower table shows pairwise statistical significance compared to the best configuration ($L = 4$).

| Context Size ($L$) | Dice | IoU | |
|---|---|---|---|
| 1 | $0.871 \pm 0.155$ | $0.792 \pm 0.181$ | |
| 2 | $0.880 \pm 0.145$ | $0.804 \pm 0.170$ | |
| 4 | $\mathbf{0.931 \pm 0.108}$ | $\mathbf{0.877 \pm 0.133}$ | |
| 8 | $0.865 \pm 0.177$ | $0.785 \pm 0.196$ | |
| 16 | $0.738 \pm 0.338$ | $0.672 \pm 0.325$ | |
| **Compared Contexts** | **Dice $p$-value** | **IoU $p$-value** | **Significance** |
| 1 vs 4 | $1.07 \times 10^{-7}$ | $1.86 \times 10^{-10}$ | Highly significant |
| 2 vs 4 | $2.83 \times 10^{-6}$ | $1.11 \times 10^{-8}$ | Highly significant |
| 4 vs 8 | $1.34 \times 10^{-7}$ | $4.12 \times 10^{-10}$ | Highly significant |
| 4 vs 16 | $6.74 \times 10^{-18}$ | $1.06 \times 10^{-20}$ | Highly significant |

Table 7 shows that segmentation performance L=4 achieves the highest Dice and IoU scores with context size $L = 4$. The statistical analysis confirms that all other configurations differ significantly from L=4, highlighting it as the optimal choice and confirming the analysis of 3.3.

## Appendix E. Evaluation of Shared Encoder-Decoder Weights for ICL-NoiseUNet

We further evaluate our approach when we use shared encoder–decoder weights for the target and context backbones. In the CAMUS (Leclerc et al., 2019) dataset, ICL-NoiseUNet reaches a Dice of 0.933. We also implement an ICL-NoiseWNet version, in which a W-Net (Xia and Kulis, 2017) with shared parameters is utilized for the target and context branch. ICL-NoiseWNet achives a Dice score of 0.940 and a Recall score of 0.966. In thyroid gland segmentation, full design variant consistently achieves 2% or higher Dice scores compared to variants without the Noise Modulation Block (NMB). Thus, it is highlighted that the integration of the NMB to each encoder-decoder block provides a consistent performance advantage across different medical segmentation tasks.

Table 8: Quantitative results for different variants of our framework, with shared weights for encoder-decoder blocks of the target input and context set.

| Model / Dataset | Dice | IoU | Recall |
|---|---|---|---|
| **CAMUS (Leclerc et al., 2019) Dataset** | | | |
| ICL-NoiseUNet (shared weights) | 0.933 | 0.890 | 0.925 |
| ICL-UNet (shared weights - Noise Modulation Block) | 0.903 | 0.822 | 0.842 |
| ICL-NoiseWNet (shared weights) | **0.940** | **0.895** | **0.966** |
| ICL-WNet (shared weights - Noise Modulation Block) | 0.936 | 0.891 | 0.960 |
| **Thyroid-Magdeburg (Wunderling et al., 2017) Dataset** | | | |
| ICL-NoiseUNet (shared weights) | 0.774 | 0.686 | 0.736 |
| ICL-UNet (shared weights - Noise Modulation Block) | 0.731 | 0.644 | 0.693 |
| ICL-NoiseWNet (shared weights) | **0.785** | **0.701** | **0.806** |
| ICL-WNet (shared weights- Noise Modulation) | 0.745 | 0.664 | 0.749 |

## Appendix F. Inference Time and Parameter Size Analysis

Table 9: Inference characteristics across segmentation models (100 samples, single GPU).

| Model | Inference time (s) | Parameters (millions) | Model size (MB) |
|---|---|---|---|
| ICL-NoiseUNet | 12.23 | 123.440 | 470.89 |
| MultiverSegNet (Wong et al., 2025) | 9.04 | 1.182 | 4.51 |
| UNet-Transformer (Hatamizadeh et al., 2021) | 9.87 | 94.837 | 361.78 |
| Swin-UNet (Cao et al., 2021) | 10.51 | 27.165 | 103.63 |
| MedSAM2 (Ma et al., 2025) | 14.02 | 46.400 | 177.00 |

## Appendix G. Effect of Different Context Sizes at Inference

To further evaluate the robustness of ICL-NoiseUNet to variations in context size, we trained the model using a fixed context size of $L = 4$ and subsequently tested it with different context sizes across datasets. We present the Dice and IoU distributions for the BUS-BRA

(Wilfrido Gómez-Flores et al., 2023) and CAMUS (Leclerc et al., 2019) datasets, respectively. The results demonstrate that segmentation performance remains stable across a wide range of context sizes. In other words, the model effectively captures contextual information during training, enabling reliable segmentation even when the available contextual information changes at inference time.

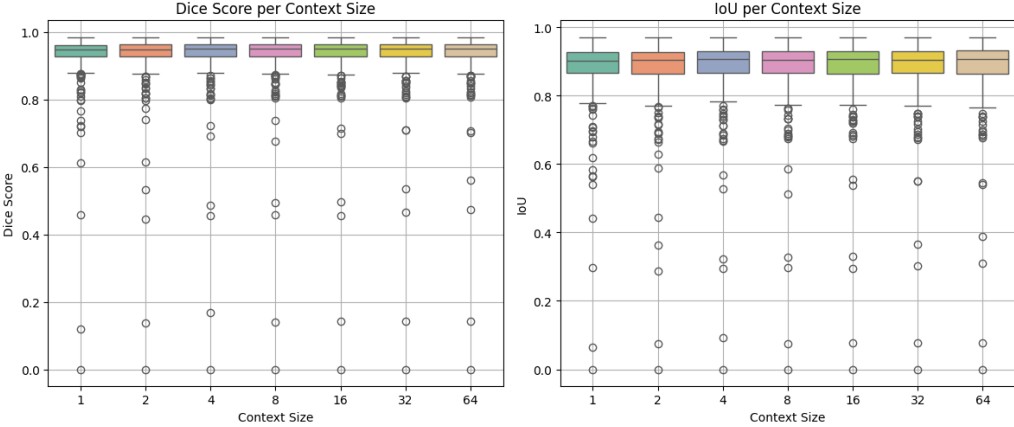

Figure 8: Effect of varying context size during **inference** on segmentation performance for the BUS-BRA (Wilfrido Gómez-Flores et al., 2023) dataset. The model was trained with a fixed context size of $L = 4$.

## Appendix H.  Representation Level Analysis

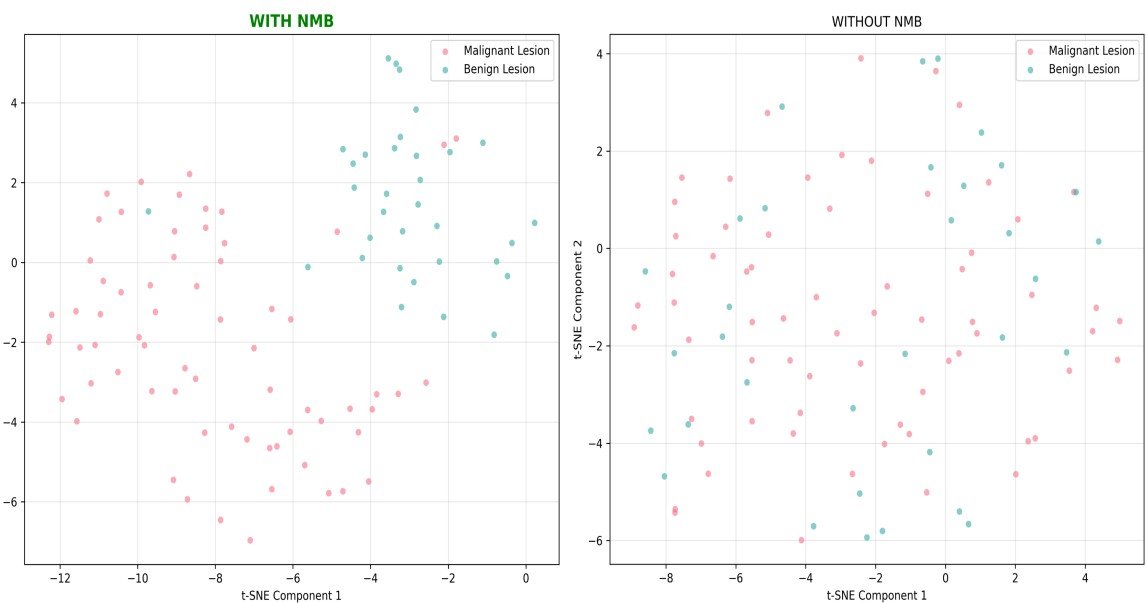

Figure 9: t-SNE visualization of bottleneck features for benign and malignant BUS-BRA (Wilfrido Gómez-Flores et al., 2023) samples. Without NMB, the learned features exhibit substantial inter-class overlap. In contrast, incorporating NMB leads to clearer class separation, as reflected by a higher silhouette score (0.26 vs. 0.05).

## Appendix I.  Robustness to Context Selection Method

| Context Pool | Selection Strategy | BUS-BRA | | CAMUS | |
|---|---|---|---|---|---|
| | | Dice | Sig. | Dice | Sig. |
| Full Set | L2 Distance | 0.911 | – | 0.940 | – |
| 10% Subset | Random | 0.910 | † | 0.936 | † |
| Full Set | SSIM | 0.902 | ‡ | 0.937 | † |

Table 10: Robustness of segmentation performance to context pool size and selection method. Minimal Dice score degradation is observed when using only 10% of the training set or different context selection metrics (L2 vs. SSIM). BUS-BRA (Wilfrido Gómez-Flores et al., 2023) results are stratified by scanner. Significance levels: † $p < 0.05$, ‡ $p < 0.01$. We compute the $p$-value for Dice score significance against the L2 distance selection strategy using Wilcoxon signed-rank pairwise tests with Bonferroni correction.

## Appendix J. Comparison with Few-Shot Learning Approaches and Feature-Wise Conditioning

| Model | CAMUS | | | BUS–BRA | | |
|---|---|---|---|---|---|---|
| | Dice ↑ | IoU ↑ | $p$–value | Dice ↑ | IoU ↑ | $p$–value |
| **ICL-NoiseUNet** | **0.940** | **0.891** | – | **0.911** | **0.831** | – |
| PANet (Wang et al., 2020) | 0.886 | 0.789 | † | 0.801 | 0.679 | † |
| FiLM (Perez et al., 2017) | 0.926 | 0.879 | † | 0.867 | 0.781 | † |
| ICL-NoiseUNet (-nm,+cbn) | 0.930 | 0.881 | † | 0.891 | 0.813 | † |

Table 11: Comparison with current State-of-The-Art few-shot learning models and feature-wise conditioning mechanisms (nm: Noise Modulation Block, cbn: Conditional BatchNorm (de Vries et al., 2017)). BUS-BRA (Wilfrido Gómez-Flores et al., 2023) results are stratified by scanner. Significance levels: † $p < 0.001$. We compute the $p$-value for Dice score significance against ICL-NoiseUNet using Wilcoxon signed-rank pairwise tests with Bonferroni correction.

## Appendix K. Learned Modulation Parameters

The learned residual weights $\alpha_k$ and variance weights $\beta_k$ exhibit clear trends across datasets, as summarized in Table 12. For the CAMUS (Leclerc et al., 2019) and Thyroid-Madgeburg (Wunderling et al., 2017) datasets, both weights start at relatively low values in the early network blocks (approximately 0.44–0.48) and progressively increase with network depth, reaching 0.55–0.56 in the deepest blocks. This indicates that the model learns to apply stronger noise modulation in deeper layers, where features become more semantic. In contrast, the BUS-BRA (Wilfrido Gómez-Flores et al., 2023) dataset shows a different behavior, with $\alpha_k$ and $\beta_k$ remaining relatively stable across layers and confined to a narrow range of 0.48–0.52. Consequently, breast ultrasound segmentation benefits from applying similar noise handling at all network blocks. Across all datasets, the residual and variance weights remain approximately equal, so the analytic descriptors contribute equally at every network level. Overall, these observations show that the proposed model adapts its noise modulation strategy to each task, rather than applying a single approach across all datasets.

Table 12: Learned residual ($\alpha_k$) and variance ($\beta_k$) weights of the NMB across network blocks for different ultrasound datasets.

| Block | BUS-BRA | | CAMUS | | Thyroid | |
|---|---|---|---|---|---|---|
| | $\alpha_k$ | $\beta_k$ | $\alpha_k$ | $\beta_k$ | $\alpha_k$ | $\beta_k$ |
| 1 | 0.480 | 0.480 | 0.444 | 0.443 | 0.453 | 0.458 |
| 2 | 0.481 | 0.483 | 0.470 | 0.473 | 0.491 | 0.488 |
| 3 | 0.493 | 0.493 | 0.491 | 0.491 | 0.492 | 0.490 |
| 4 | 0.499 | 0.497 | 0.518 | 0.504 | 0.511 | 0.501 |
| 5 | 0.501 | 0.500 | 0.535 | 0.514 | 0.519 | 0.513 |
| 6 | 0.501 | 0.498 | 0.539 | 0.518 | 0.514 | 0.510 |
| 7 | 0.502 | 0.499 | 0.535 | 0.511 | 0.528 | 0.528 |
| 8 | 0.521 | 0.519 | 0.546 | 0.547 | 0.533 | 0.532 |
| 9 | 0.524 | 0.523 | 0.558 | 0.551 | 0.553 | 0.558 |

