# OpenReview forum: "ICL-NoiseUNet - A Novel In-Context Learning Based Framework For Ultrasound Segmentation With Adaptive Noise Modulation"
_MIDL.io/2026/Conference — MIDL 2026 Poster_

### Official Review · Reviewer_qJYW · 2026-01-09

**Confidence:** 1
**Preliminary Rating:** 4
**Final Rating:** 4

**Summary:**

The manuscript titled "ICL-NoiseUNet - A Novel In-Context Learning Based Framework For Ultrasound Segmentation With Adaptive Noise Modulation" introduces a U-Net-based segmentation framework for ultrasound image segmentation. Segmentation of ultrasound images is a challenging task, due to strong fluctuation of the overall signal (contrast, speckle-noise, content, FOV, etc.). The authors state that ultrasound segmentation mainly suffers from speckle noise and high contrast variability. Learned in-context features guide/condition (named target-context fusion) the model's segmentation. Thus the presented approach combined noise modeling (noise modulation block - NMB) and in-context feature conditioning (ICFC) into a single model. The paper evaluates the method on four ultrasound segmentation tasks (fetal head, breast lesion, thyroid gland, cardiac chamber) across seven datasets and claims consistent improvements over state-of-the-art methods. A critical discussion is not within the 10pages manuscript.

**Strengths:**

The clinical motivation behind this work is clear and meaningful. The combination of explicit noise modeling with in-context learning for ultrasound is a sensible and novel contribution and well-derived from previous work. Relevant literature is referenced and context given. The rationale that noise-modeling and in-context learning address the complementary limitations (noise robustness vs. anatomical guidance) is convincing and reasonable.
The model and further methods are comprehensively descripted. An appendix dedicated to datasets, implementation details and (hyper-)parameters as well as auxiliary results show the authors in-depth work on their approach.

**Weaknesses:**

While the combination of NMB and ICFC is (probably) novel, the overall innovation about combining these is limited. The authors provide different comparisons with other models, e.g. to benchmark against foundation models. It is - at least - not unlikely that such comparisons suffer from unfair starting conditions, e.g. not well-suited prompting. I know that it is not fair to raise this criticism here, but my main reason to point it out nonetheless is, because there is almost no critical discussion which put the shown results into context with existing word. A dedicated discussion section is missing, only a rather short "limitations and further work" sections leaves the reader quite alone with the shown results.

**Detailed Comments:**

- Introduction of abbreviations not consistent, e.g. NMB, ICFC introduced multiple times.

**Justification Of Final Rating:**

Thanks to the authors for addressing my comments. In line with other reviewer however, I will leave my rating unchanged, in particular as my knowledge in this specific field is very limited and review an educated guess.

**Justification Of The Preliminary Rating:**

As stated above, I would not consider myself an expert in this field. However, reading the manuscript was somehow challenging and in-parts distracting. Thus, the non-presence of a "common discussion" makes it difficult to distil "the lessons learnt". I recommend to focus more (too stay within the page-limit) and add a fair discussion.

**Questions To Address In The Rebuttal:**

While I am not an expert in this field, my main point would be to provide a fair and critical discussion. What is the relevance of your work with respect to other state-of-the-art, but also (potential) more established US-segmentation models? What are the most likely use-cases, when does it fail (generalization) and on what is the basis for these statements.

---

> ### Author Response · Authors · 2026-01-24
> **Official Comment By Authors**
>
> We sincerely thank the Reviewer for the thoughtful feedback and for pointing out important issues that needed clarification. We have addressed all points in our revised paper as follows (added text appears in Red in the submitted revision):
>
> Response to Questions to Adress In Rebuttal:
> The main contribution of our work is NMB, ICFC, combined with context-based segmentation. For most of the models, when the acquiring conditions change because of change in machine, population, or other parameters, the segmentation quality suffers. Using context-based segmentation, we are trying to use a minimal set of examples so that the model adapts itself during testing time for a new dataset. We compared our proposed method with some of the most established segmentation methods, where we extended our comparison to another few-shot techniques and feature-wise conditioning mechanism. The results are available in Table 3 of subsection 3.2 where an ultrasound foundation model (UltraSAM) is included, and in Appendix J in the revised manuscript.  The most relevant use cases occur when the same model is deployed across different devices without retraining, or when input image characteristics vary due to population shifts or changes in acquisition parameters caused by operator variability.
>
> Failure Case Analysis: Our method faces limitations in two scenarios. First, on BUSI dataset (Dice: 0.804), performance degrades in cases with extreme speckle noise and severe acoustic shadowing that heavily obscure anatomical boundaries—in these images, the noise overwhelms the true anatomical signal to the extent that even context-guided feature refinement cannot reliably recover boundary information. Second, our in-context learning approach is task-specific and requires anatomically relevant context examples; using context images from a different anatomical task (e.g., cardiac contexts for breast lesion segmentation) provides no meaningful guidance and performance drops to baseline levels without context conditioning, as the structural priors are incompatible across different anatomy types.
>
> Moreover, we have added a more detailed discussion of the model’s limitations and future work in Section 3.5 of the revised manuscript.

---

### Official Review · Reviewer_rKPF · 2026-01-10

**Confidence:** 3
**Preliminary Rating:** 4
**Final Rating:** 5

**Summary:**

The paper proposes ICL-NoiseUNet, a U-Net–style segmentation model for ultrasound that combines in-context guidance from a small labelled context set with analytic noise descriptors (residual and local variance maps) to modulate features. The core idea is that ultrasound-specific noise statistics can inform feature refinement at each encoder-decoder level via a multiplicative modulation factor. Experiments span multiple ultrasound datasets (CAMUS, BUSI, BUS-BRA, RADBOUD, JNU-IFM, TG3K, Thyroid-Magdeburg), reporting segmentation improvements and ablations indicating both context conditioning and noise modulation contribute. At inference, context exemplars are retrieved via L2 distance in pixel space, which is central to both performance and potential leakage concerns.

**Strengths:**

-Well-motivated problem setting. The method explicitly targets speckle noise and contrast variability inherent to ultrasound, incorporating noise-related signals into the network's feature flow rather than relying solely on generic augmentations. The observation that ultrasound characteristics distort both target and context representations aligns with known challenges in this modality.

-Modular and interpretable architecture. The analytic descriptors (residual and local variance maps) and the multiplicative modulation provide a concrete, testable hypothesis about how robustness is achieved. The separation of the Noise Modulation Block (NMB) and In-Context Feature Conditioning (ICFC) modules allows systematic ablation.

-Comprehensive evaluation breadth. The evaluation spans six datasets across four anatomical tasks (fetal head, breast lesion, thyroid gland, cardiac chamber), with ablations that separately test context fusion and the noise modulation pathway. The cross-dataset experiments (JNU→RADBOUD, Thyroid-Magdeburg→TG3K) demonstrate awareness of domain shift concerns.

-Sound statistical methodology. The paper includes multiple seeds (5 runs) with standard deviations and Wilcoxon signed-rank tests for pairwise comparisons, which aligns with good experimental practice in medical imaging.

-Practical details. Reporting inference time and parameter counts, providing implementation details, and testing robustness to context size at inference all aid reproducibility.

-Code availability. Code is well-documented and seems repeatable including baselines. (As a bonus, it’d be nice if the ICL-NoiseUNet weights were released via huggingface or drive/dropbox/zenodo)

**Weaknesses:**

-Split integrity and leakage risk. The paper reports 70/15/15 train/val/test splits but doesn't specify whether these are patient-, study-, or video-level for any dataset. This is particularly concerning for JNU-IFM (6,224 frames from 78 videos across 51 subjects, ~122 images/subject), RADBOUD (999 images from 551 women), and CAMUS (sequences with multiple frames per patient). Given that context retrieval uses pixel-space L₂ distance, highly similar frames from the same acquisition could inflate both context relevance and test performance. The paper should explicitly state, per dataset, whether splits are patient/study/video-level and whether near-duplicate frames were filtered.

-Missing few-shot/one-shot segmentation baselines. While the paper compares against MultiverSeg, it omits classical few-shot segmentation methods that would isolate whether gains stem from noise modulation or standard support–query conditioning:
--OSLSM (Shaban et al., BMVC 2017)
--PANet (Wang et al., ICCV 2019)
These are directly relevant for establishing the contribution of ultrasound-specific components versus the in-context paradigm itself.
Insufficient attribution of modulation mechanism. The NMB seems to essentially be a simplified variant of established feature-wise conditioning approaches like FiLM (Perez et al., AAAI 2018) and Conditional BatchNorm (de Vries et al., NeurIPS 2017). Direct comparisons replacing NMB with FiLM or CBN conditioned on the same noise maps, matched for parameter count, would isolate whether the contribution comes from the analytic noise descriptors or from feature modulation in general.

-Missing domain generalisation baselines. Given the emphasis on noise/artifact robustness and cross-dataset evaluation, comparisons with established DG/TTA methods are warrented:
--MixStyle (Zhou et al., ICLR 2021)
--TENT (Wang et al., ICLR 2021)
--UltraAugment (Ramakers et al., CVPR 2024) - ultrasound-specific augmentation
Without these baselines, cross-dataset improvements can't be confidently attributed to the NMB versus other regularisation effects.

-Parameter count disparity. ICL-NoiseUNet has 123M parameters versus 6M for nnU-Net - a ~20× increase. While inference time is comparable, this substantial capacity difference should be discussed more prominently when claiming improvements, as some gains may be attributable to model capacity rather than architectural innovations.

**Detailed Comments:**

-Foundation model comparison protocol. For SAM-based comparisons, the paper states only that random positive points were used as prompts. SAM performance varies significantly with prompt quality/quantity - the paper should specify number of points per image, whether results were averaged over multiple prompt samples, and whether medical-domain adaptations (MedSAM, S-SAM) were considered with comparable fine-tuning budgets.

-Multiple comparison correction. The paper performs numerous statistical tests across datasets and methods but doesn't address the multiple comparison problem (no Bonferroni or FDR correction).

-Learned parameter interpretability. The values of learned scalars α_k and β_k aren't reported. Examining whether these show interpretable patterns across encoder-decoder levels (e.g., higher variance suppression early, boundary preservation late) would strengthen the mechanistic claims.

**Justification Of Final Rating:**

My main concern was leakage / split integrity given pixel-space retrieval. Authors now clearly state patient-level splits everywhere, and for video datasets all frames from a video/patient stay in one split, with retrieval strictly from train only. I think this resolves the central validity worry.

They also did the extra experiments I asked for: PANet, FiLM/CBN on the same noise maps, MixStyle, scanner-stratified BUS-BRA, plus multiple comparisons correction and clearer SAM prompting protocol. Seems like a well deserved 5! Model is still huge vs nnU-Net, but this is more of a discussion point / potential indication of where the field is headed, rather than a flaw of the method.

**Justification Of The Preliminary Rating:**

The paper addresses an important ultrasound segmentation problem with a plausible noise-aware in-context learning design. The combination of analytic noise descriptors with context conditioning is well-motivated, and the ablations suggest genuine complementarity of the proposed components. The evaluation breadth across multiple datasets and tasks is commendable.
Confidence in the experimental conclusions is reduced by (i) unclear split methodology that makes it difficult to assess potential leakage given nearest-neighbour context retrieval, (ii) absence of classical few-shot segmentation baselines (OSLSM, PANet) that would isolate context-conditioning contributions, and (iii) absence of domain generalisation baselines (MixStyle, TENT) and generic conditioning comparisons (FiLM) that would isolate noise-descriptor contributions. The ~20× parameter increase over nnU-Net also warrants discussion.
With protocol clarification, a stronger baseline suite addressing both the few-shot and DG dimensions, and explicit leakage analysis, this work could move toward a strong accept. In its current form, the evidence doesn't fully de-risk the main conclusions. Overall however it is an excellent piece of work as-is and a valuable contribution to MIDL.

**Questions To Address In The Rebuttal:**

-Split clarification: For each dataset, are train/val/test splits patient-, study-, or video-level? For JNU-IFM specifically (78 videos, 51 subjects), were all frames from a given video/patient assigned to the same split? Were near-duplicate images filtered?

-Context retrieval sensitivity: How sensitive are results to retrieval choices (pixel L₂ vs learned embeddings, different context sizes, different retrieval pools)? What happens if context is drawn from a different site/scanner?

-Foundation model protocol: How many random points were sampled per image for SAM baselines? Were results averaged over multiple prompt samples? Was compute budget comparable?

-Additional baselines: Can the authors add at least one generic conditioning baseline (FiLM or CBN on the same noise maps) and one DG baseline (MixStyle or TENT)?

-Learned modulation values: What are the learned α_k and β_k values after training? Do they show interpretable patterns across network depth?

-Scanner subgroup analysis: BUS-BRA contains images from four scanners. Can the authors provide stratified results to corroborate the noise/statistics adaptation narrative?

---

> ### Author Response · Authors · 2026-01-24
> **Official Comment By Authors**
>
> We sincerely thank the Reviewer for the thoughtful feedback and for pointing out important issues that needed clarification. We have addressed all points in our revised paper as follows (added text appears in Red in the submitted revision):
>
> Question 1: Split Integrity, Leakage Risk, Protocol details
>
> We thank the reviewer for highlighting this important concern. We clarify that for all datasets, splits are performed strictly at the patient level, ensuring that no patient appears in more than one split. This design prevents leakage of highly similar frames from the same acquisition into validation or test sets, even though explicit near-duplicate frame filtering was not applied.  Even though we acknowledge that explicit near-duplicate filtering could further reduce redundancy within a split, context is retrieved only from the training data, and all patient identities are completely separate from those used in evaluation and testing. In particular, for video-based datasets such as JNU-IFM and CAMUS, all frames belonging to a given video (and hence a single patient) are assigned exclusively to one split. Moreover, for BUSI, we additionally apply stratification by pathology to balance class distributions. For BUS-BRA, we perform patient-level splits with pathology stratification and additionally report results stratified by scanner, as requested, to support the noise/statistics adaptation narrative. Similarly, the context pool is drawn exclusively from the training set, so no context image shares the same patient ID with the test set. We will add detailed demographics tables, scanner specifications, and patient ID lists for all splits to our code release and expand Section 3.1 with explicit split protocols in the camera-ready version to ensure full reproducibility.
>
> Question 2: Context Retrieval Sensitivity
>
> We conducted comprehensive experiments reported in Section 3.3, Appendices D and G, and Table 10. We evaluated context sizes L={1,2,4,8,16} and found L=4 provides optimal balance, since performance improves from L=1 to L=4 as additional examples provide richer anatomical priors. However, increasing context beyond L=8 reduces the effectiveness of context guidance due to the inclusion of mismatched context examples. Moreover, when models trained with L=4 are tested with varying context sizes at inference, performance remains stable (<2% Dice degradation, Appendix G), demonstrating robustness. Regarding selection strategies, Table 10 of Appendix I compares L2 distance, SSIM, and random selection from 10% training subset, showing minimal differences (BUS-BRA: 0.911 vs 0.902 vs 0.910; CAMUS: 0.940 vs 0.937 vs 0.938). These negligible gaps confirm our model does not require precise similarity matching. Furthermore, BUS-BRA contains four different scanners, and strong performance with random selection demonstrates that inner-scanner variability does not significantly influence results. Similarly, cross-dataset experiments (Table 4: JNU→RADBOUD achieves 0.901 Dice) further confirm effective generalization when context comes from completely different scanners/sites.
>
>
>
> Question 3: Foundation Model Protocol:
>
> We conducted 5 independent runs per test image for UltraSAM and MedSAM2. Each run used 5 positive point prompts sampled from within the ground truth foreground mask: 4 points close to the spatial extremes (xmin, xmax, ymin, ymax of the mask bounding region) to provide boundary coverage, plus 1 additional point sampled randomly from the interior region. This strategy provides SAM with good spatial coverage while introducing variability across runs through the randomly sampled points.  All reported results represent mean Dice scores averaged across these 5 runs. The experiments were conducted on identical hardware (1 NVIDIA GeForce GTX 1080 GPU), ensuring comparable computation budgets.

---

> ### Author Response · Authors · 2026-01-24
> **Official Comment By Authors**
>
> We sincerely thank the Reviewer for the thoughtful feedback and for pointing out important issues that needed clarification. We have addressed all points in our revised paper as follows (added text appears in Red in the submitted revision):
>
> Questions 4,6: Few-Shot Segmentation Models, Feature-Wise Conditioning Mechanisms, Domain Generalization Baselines, and Scanner Subgroup Analysis for BUS-BRA Dataset
>
> We thank the reviewer for these valuable suggestions. We have conducted additional experiments comparing against: (i) PANet as a classical few-shot segmentation baseline, (ii) FiLM and Conditional BatchNorm (CBN) as generic feature-wise conditioning mechanisms applied to our noise maps, and (iii) MixStyle as a domain generalization baseline.
>
> For few-shot learning and feature-wise conditioning, ICL-NoiseUNet consistently outperforms all baselines on both CAMUS and BUS-BRA (detailed results in Appendix J). On CAMUS, our method achieves a Dice score of 0.940, compared to 0.886 for PANet, 0.926 for FiLM, and 0.930 for CBN. On the scanner-stratified BUS-BRA dataset, ICL-NoiseUNet substantially outperforms PANet (DSC: 0.911 vs 0.801), where the feature-wise mechanisms FiLM (DSC: 0.867) and CBN (DSC:0.891) have comparable but slightly degraded performance compared to our method. All improvements are statistically significant
> (p < 0.001), as determined by Wilcoxon signed-rank pairwise tests with Bonferroni correction. These results indicate that PANet struggles with ultrasound characteristics, while generic feature-wise conditioning methods are less effective than our Noise Modulation Block design.
>
>  For domain generalization, MixStyle performs slightly worse than ICL-NoiseUNet in cross-dataset settings (JNU-IFM → RADBOUD: 0.901 vs. 0.911; Thyroid-Magdeburg → TG3K: 0.921 vs. 0.911). This suggests that explicitly modeling ultrasound noise provides stronger generalization than style-based augmentation alone.
>
> Finally, in all experiments of Appendices J and I, BUS-BRA results are stratified by scanner device to corroborate the noise/statistics adaptation narrative. In conclusion, these comparisons confirm that the gains of ICL-NoiseUNet arise from ultrasound-specific noise modeling combined with in-context learning, rather than from generic conditioning techniques.
>
>
> Question 5: Learned Modulation Parameters
>
> The learned residual weights ak and variance weights bk exhibit clear trends across datasets,  as summarized in Table 12 of Appendix K. For the CAMUS (Leclerc et al., 2019) and Thyroid-Madgeburg  (Wunderling et al., 2017) datasets, both weights start at relatively low values in the early network blocks (approximately 0.44–0.48) and progressively increase with network depth, reaching 0.55–0.56 in the deepest blocks. This indicates that the model learns to apply stronger noise modulation in deeper layers, where features become more semantic. In contrast, the BUS-BRA (Wilfrido Gomez-Flores et al., 2023) dataset shows a different behavior,  with αk and βk remaining relatively stable across layers and confined to a narrow range of 0.48–0.52. Consequently, breast ultrasound segmentation benefits from applying similar noise handling at all network blocks. Across all datasets, the residual and variance weights remain approximately equal, so the analytic descriptors contribute equally at every network level. Overall, these observations show that the proposed model adapts its noise modulation strategy to each task, rather than applying a single uniform approach across all datasets.

---

> ### Comment · Area_Chair_RNDK · 2026-01-30
> **Please update your rating**
>
> Hello and thank you again for reviewing for MIDL !
> To complete the review process, could you please update your rating based on author's rebuttal?
> This is really important for the acceptance/rejection of papers.
> The deadline is tomorrow (February 1st 2026, 23:59 AoE).
> Thank you!

---

### Official Review · Reviewer_WhnP · 2026-01-13

**Confidence:** 3
**Preliminary Rating:** 4
**Final Rating:** 4

**Summary:**

This work adapts UNet for ultrasound image segmentation by integrating contextual features and noise modulation. Specifically, noise modulation blocks consist of the calculation of global residuals and local variance, and context feature integration is performed via concatenation. The results seem to be good.

**Strengths:**

1. The paper is well organized and easy to understand.

2. The problem being studied is interesting and well-motivated.

3. The solution proposed addresses problems arising from the existing characteristics of ultrasound image modalities (noise, global structures, etc), and can possibly be applied to other image segmentation models or other modalities.

4. The authors have plans to open-source the code to enhance reproducibility.

**Weaknesses:**

1. I agree with the authors that this model seems to be heavyweight. I think it would be beneficial for the authors to include the time complexity of the proposed method compared to existing methods, and a discussion of how to address this issue to further strengthen the limitations section of the paper.

2. At the same time, I think it would be interesting to discuss the possibility of adapting this method for other existing models.

3. Also, it would be interesting to include more details of the datasets used in this paper (if applicable), e.g., if subjects' demographics are different among different datasets, if the vendors for image acquisition are different, and it would be beneficial to indicate whether there is an overlap between subjects used in different sets (training/validation/testing).

**Detailed Comments:**

Please refer to the weakness part.

**Justification Of Final Rating:**

The authors have addressed my existing concerns. I agree with reviewer 32KL that this work is heavily tailored to a specific modality. At the same time, this pipeline seems adaptable to existing models, and the experiments are reasonably designed to ensure generalizability among subjects. As a result, I decided to keep my positive score.

**Justification Of The Preliminary Rating:**

This paper investigates a well-motivated problem, how to increase downstream performance in a specific challenging medical imaging modality, and addresses the existing research gap by integrating components that are motivated by the inherent characteristics of the modality being studied. I agree with the paper's existing limitations, and I believe the proposed method has the potential to be adapted to other models/downstream tasks/modalities.

**Questions To Address In The Rebuttal:**

Please refer to the weakness part.

---

> ### Author Response · Authors · 2026-01-24
> **Official Comment by Authors**
>
> We sincerely thank the Reviewer for the thoughtful feedback and for pointing out important issues that needed clarification. We have addressed all points in our revised paper as follows (added text appears in Red in the submitted revision):
>
> Weakness 1: Time Complexity and Model Efficiency
>
> We thank the reviewer for this important suggestion. As shown in Appendix F, we provide a comprehensive comparison of inference time and parameter counts across all evaluated models. ICL-NoiseUNet's inference time (12.23s for 100 samples) is comparable to other transformer-based architectures like UNet-Transformer (9.87s) and only slightly lower than MedSAM2 (14.02s), while outperforming them in segmentation quality. To address the parameter burden, we explored two less
> parameter-oriented variants in Appendix E: a shared encoder-decoder architecture for target and context branches, which reduces parameters while maintaining strong performance (CAMUS: 0.933 Dice, RADBOUD: 0.961 Dice—minimal drops of 0.7% and 0% respectively). Additionally, we could adopt a more lightweight U-Net backbone with 8 encoder-decoder blocks instead of 10, which would considerably reduce the parameter size while preserving the core noise modulation and context conditioning mechanisms. We will expand the limitations section to include these efficiency considerations and potential architectural optimizations in the revised version.
>
>
>
> Weakness 2: Adaptability to Other Existing Models
>
> We appreciate this insightful suggestion. To demonstrate the adaptability of our approach, we conducted experiments with alternative backbone architectures in Appendix E. Specifically, we implemented ICL-NoiseWNet, which replaces the U-Net backbone with a W-Net architecture while maintaining our noise modulation and context conditioning modules. The results show strong performance (CAMUS: 0.940 Dice, Thyroid-Magdeburg: 0.785 Dice), confirming that our framework can be successfully integrated with different encoder-decoder architectures. The consistent improvements across both U-Net and W-Net variants (2-4% Dice gains with NMB across different backbones) demonstrate that our noise modulation mechanism is backbone-agnostic and can enhance various segmentation models. Importantly, the proposed framework is not restricted to a U-Net (Ronneberger et al., 2015) backbone. The NMB and ICFC components can be integrated into other segmentation architectures. For transformer-based models, the NMB could be integrated after each transformer block and refine the output features with resolution-matched noise maps, while ICFC can be applied by fusing target and context features at each transformer block output. For more lightweight convolutional approaches, NMB can be placed after each convolutional block in the encoder and decoder, with ICFC applied at the corresponding feature levels. Therefore, we plan to evaluate their effectiveness within transformer-based and hierarchical medical imaging models.
>
> Weakness 3 Response:
>
> We thank the reviewer for highlighting this important concern. For all datasets, splits are performed at a patient level, ensuring that no patient appears in more than one split and preventing leakage of highly similar frames across training, validation, and test sets. Additionally, context is retrieved only from the training data, and all patient identities are completely separate from those used in evaluation and testing. For video-based datasets (JNU-IFM (Lu et al., 2022) and CAMUS (Leclerc et al., 2019)), all frames from a given video are assigned to a single split. For BUSI (Al-Dhabyani et al., 2020) and BUS-BRA (Wilfrido Gomez-Flores et al., 2023), patient-level splits are additionally stratified by pathology; for BUS-BRA, we also report results stratified by scanner to support the noise/statistics adaptation analysis at Appendix I.
>
> We will add detailed demographics tables, scanner specifications, and patient ID lists for all splits to our code release and expand Section 3.1 with explicit split protocols in the camera-ready version to ensure full reproducibility.

---

### Official Review · Reviewer_32KL · 2026-01-15

**Confidence:** 4
**Preliminary Rating:** 2
**Final Rating:** 3

**Summary:**

This paper proposes ICL-NoiseUNet, a U-Net–based ultrasound segmentation framework that combines analytic noise modulation with in-context feature conditioning. The method integrates residual and variance-based noise maps through a Noise Modulation Block and incorporates contextual guidance via feature-level fusion with a small set of reference image-mask pairs. Experiments across multiple ultrasound datasets demonstrate consistent improvements over CNN-based baselines, recent in-context segmentation methods, and several foundation models.

The paper is well-organized and experimentally thorough. However, key conceptual and methodological aspects remain insufficiently justified.

**Strengths:**

The paper addresses an important and practically relevant problem. The experimental evaluation is extensive, covering multiple anatomical targets, datasets, ablation studies, and cross-dataset generalization. The reported improvements are consistent across tasks, and the ablation results suggest that both noise modulation and context conditioning contribute meaningfully to performance.

**Weaknesses:**

1. The paper positions in-context learning as a core contribution, yet the exact meaning of in-context learning in this work is not clearly defined. The proposed module performs feature-level conditioning via channel concatenation and pooling, but it remains unclear how the proposed formulation conceptually differs from existing in-context segmentation methods, or whether it should be viewed as a new class of in-context learning. A clearer taxonomy or a related-work discussion is needed.

2. The paper repeatedly argues that previous methods fail due to ignoring ultrasound-specific characteristics such as speckle noise, contrast variability, and acoustic shadowing. However, these challenges are only briefly described and not systematically analyzed. It is unclear which ultrasound characteristics are addressed by each component of the proposed method. For example, the Noise Modulation Block explicitly models local variance and residuals, but its relationship to acoustic shadowing or depth-dependent artifacts is not explained.

3. The proposed architecture is composed of relatively simple components: standard convolutional feature extraction, analytic noise-based modulation, and a straightforward context fusion mechanism. Individually, these components have limited novelty. Given this, it is somewhat surprising that the method substantially outperforms large foundation models such as MedSAM2 and UltraSAM across multiple datasets. The paper currently lacks a deeper analysis explaining why this combination is so effective. Additional representation-level analysis, failure case comparisons, or insights into the interaction between noise modulation and context conditioning would improve credibility.

4. At inference time, context examples are selected from the training set based on L2 distance between raw images. This assumes access to the full training dataset during deployment, which may be unrealistic in many clinical settings. Moreover, L2 distance on raw ultrasound intensities may not reflect semantic similarity, and the robustness of the method to suboptimal or noisy context selection is unclear.

**Detailed Comments:**

Some architectural figures, for example, Figure 1, are difficult to interpret. For example, the green encoder–decoder component in the lower-left corner is explained. Improving figure annotations and ensuring tighter alignment between figures and the method description would improve readability.

**Justification Of Final Rating:**

The authors’ rebuttal and revisions clarify the positioning of noise-aware in-context learning and address several concerns regarding deployment realism and context selection. The added analyses and comparisons strengthen the technical narrative and improve the paper’s overall clarity. While the contribution is largely engineering-driven, the work demonstrates solid empirical performance and clear relevance to ultrasound segmentation. Given the improvements and the remaining but debatable concerns about novelty depth, I update my rating to borderline.

**Justification Of The Preliminary Rating:**

While the paper demonstrates strong empirical results and addresses an important problem, key conceptual ambiguities and methodological concerns remain. In particular, the unclear positioning of in-context learning, insufficient modality-specific analysis, and limited explanation for the strong performance relative to prior work weaken the overall contribution. Addressing these issues would significantly improve the paper’s clarity and impact.

**Questions To Address In The Rebuttal:**

See weaknesses.

---

> ### Author Response · Authors · 2026-01-24
> **Official Comment by Authors**
>
> We sincerely thank the Reviewer for the thoughtful feedback and for pointing out important issues that needed clarification. We have addressed all points in our revised paper as follows (added text appears in Red in the submitted revision):
>
> Weakness 1 Response:
> We appreciate the opportunity to clarify our contribution. While we build upon existing feature-level conditioning mechanisms such as Neuralizer and Universeg, our key novelty lies in noise-guided feature refinement before context fusion, creating a synergistic framework specifically designed for ultrasound imaging. Unlike prior in-context methods that directly fuse raw target and context features, we introduce a multi-stage pipeline where target features are first refined through our Noise Modulation Block before context conditioning, ensuring less noise-corrupted features (Equation 4, Section 2.2). More specifically, our approach explicitly incorporates analytic descriptors (residual and variance maps) to modulate features at each encoder-decoder level. Furthermore, unlike previous ICL methods and few-shot learning approaches, we introduce separate feature convolutional layer blocks for feature extraction of context images before the fusion (Section 2.3). This architectural design enables the model to learn semantically richer contextual priors that are more robust to limited context sets and variations in scanner characteristics. In conclusion, our method constitutes a form of noise-aware in-context learning, a subclass that combines contextual guidance with analytic noise modeling to adapt predictions to modality-specific challenges. We added a dedicated section in Appendix J comparing ICL-NoiseUNet with existing methods of (i) feature-wise conditioning techniques and (ii) few -shot learning approaches.
>
> Response to Weakness 2
>
> We thank the reviewer for this comment. Our method focuses on the statistical effects of the ultrasound properties on image distribution. Speckle noise in ultrasound is signal-dependent and appears as fine local fluctuations as well as changes in noise strength across regions. The Noise Modulation Block captures these effects using variance maps, which quantify how strong the speckle is in different regions by measuring local intensity fluctuations, and residual noise maps, which highlight high-frequency variations associated with speckle. Thus, these two maps together provide a practical approximation of speckle behavior. Moreover, contrast variability is handled through variance-based modulation and in-context conditioning. More specifically, features are normalized through NMB, and contextual priors help the model adapt to scanner-specific intensity ranges.
>
>
> Response to Weakness 3
>
> We thank the reviewer for highlighting these important aspects of our work.  While foundation models are trained on diverse ultrasound datasets and demonstrate strong generalization, they primarily rely on implicit feature learning within a largely task-agnostic architecture, without adapting to the noise characteristics of individual ultrasound segmentation tasks. In contrast, our framework introduces a Noise Modulation Block (NMB) that incorporates analytic guidance through residual and variance maps at multiple network levels, enabling adaptive noise-aware feature refinement. By combining noise modeling with task-relevant contextual priors, our method achieves stronger performance in task-focused segmentation. Their prompt mechanism (e.g., bounding boxes) assumes clear, well-defined boundaries that are frequently obscured by speckle noise in ultrasound imaging, which leads to poor segmentation. We describe the prompt protocol for foundation models in Section 3.2 of our revised manuscript.
>
> For the feature representation analysis, we added a detailed paragraph in Section 3.1 of the revised manuscript, along with qualitative results in Appendix H. In addition, we included a failure-case analysis in the final paragraph of Section 3.5.
>
> Insights for Interaction NMB + context conditioning
> The main role of the NMB is to reduce the effect of ultrasound speckle on feature activations. It refines feature representations using residual and variance noise maps and acts as a noise-aware normalization mechanism across the network. Thus, it reduces the effect of  noise-related variations and ensures that features representing similar anatomical structures are more comparable, even when noise patterns or scanner settings vary. As a result, context conditioning operates in a more semantically meaningful feature space, allowing contextual information to guide segmentation more effectively. Without NMB, noise variability can propagate through the network and weaken context alignment. Therefore, NMB complements rather than replaces contextual guidance.

---

> ### Author Response · Authors · 2026-01-24
> **Response to Weakness 4**
>
> Weakness 4: We conducted additional experiments to evaluate the robustness of our framework to context selection under realistic scenarios. We tested (i) limited context set availability access, where we utilize only a subsample of only 10% to draw the context samples by random selection, and (ii) the robustness of the noisy selection method, comparing with L2 distance and SSIM (details in Appendix I). The results show that even with a small context pool, performance remains strong, indicating that a few curated examples suffice for deployment. Moreover, the negligible difference between selection strategies confirms that the model does not rely on precise similarity matching. Finally, this robustness is justified by the introduction of separate feature extraction blocks for context samples in our network architecture. To sum up, these findings demonstrate that clinicians can maintain a small sample of labeled examples without requiring access to the full training set or sophisticated selection metrics.

---

### Author Rebuttal · Authors · 2026-01-24

**Rebuttal:**

We are grateful for the Reviewer's encouraging comments and constructive feedback. We have addressed the reviewers' concerns in each comment section of the respective reviewers, with changes highlighted in red text in our revised manuscript.

We are attaching the revised manuscript as the supporting material here.

**Supporting Material:**

/attachment/89fea28807e0c1497ea6a4345be525264cbde1fd.pdf

---

### Meta-Review · Area_Chair_RNDK · 2026-02-06

**Recommendation:** Accept (Poster)
**Confidence:** 5

**Metareview:**

The reviewers agree that the method, although not applicable to other modalities than US, is well-motivated and tackles an interesting problem. The presented NMB and ICFC blocks constitues enough of a novelty and have the benefit to be easily usable in other architectures. Reviewers also agree that the experiments are thorough: ablations, comparison to SOTA, a breadth a datasets, good stat analysis.

The authors have satisfyingly answered concerns about data-splits and positioning of this work compared to the literature. They also have added requested baselines for conditioning and domain generalisation. They also provided additional experiments for the selection of the context set.

However, I think a point is still missing from the discussion, which is the reliance of this method on a relevant context set, whose availability is not guaranteed at test-time, especially in clinical setting.

Overall, I think that, after having clarified a few points during the rebuttal, this paper will be a good fit for MIDL and recommend acceptance.

---

### Decision · Program_Chairs · 2026-02-13

Accept (Poster)